# Facial EMG Activity Is Associated with Hedonic Experiences but Not Nutritional Values While Viewing Food Images

**DOI:** 10.3390/nu13010011

**Published:** 2020-12-22

**Authors:** Wataru Sato, Sakiko Yoshikawa, Tohru Fushiki

**Affiliations:** 1Psychological Process Team, Robotics Project, Baton Zone Program, RIKEN, Soraku-gun, Kyoto 619-0288, Japan; 2Field Science Education and Research Center, Kyoto University, Kyoto 606-8502, Japan; yoshikawa.sakiko.4n@kyoto-u.ac.jp; 3Kyoto University of the Arts, 2-116 Uryuyama Kitashirakawa, Sakyo, Kyoto 606-8271, Japan; 4Faculty of Agriculture, Ryukoku University, Ohtsu 520-2194, Japan; tfushiki@agr.ryukoku.ac.jp

**Keywords:** facial electromyography (EMG), food, liking, nutrition, valence

## Abstract

The physiological correlates of hedonic/emotional experiences to visual food stimuli are of theoretical and practical interest. Previous psychophysiological studies have shown that facial electromyography (EMG) signals were related to subjective hedonic ratings in response to food images. However, because other data showed positive correlations between hedonic ratings and objective nutritional values of food, whether the facial EMG reactions to food images could reflect the hedonic evaluation or nutritional assessment of food remains unknown. To address this issue, we measured subjective hedonic ratings (liking, wanting, valence, and arousal) and physiological signals (facial EMG of the corrugator supercilii, zygomatic major, masseter, and suprahyoid muscles, skin potential responses, and heart rates) while participants observed food images that had objective nutritional information (caloric, carbohydrate, fat, and protein contents). The results revealed that zygomatic major EMG activity was positively correlated with ratings of liking, wanting, and valence, but not with any objective nutritional value. These data indicate that facial EMG signals in response to food images reflect subjective hedonic experiences, but not objective nutritional values, associated with the food item.

## 1. Introduction

Hedonic responses (e.g., liking) to visual food stimuli play a crucial role in our survival and wellbeing, because “we eat first with our eyes”, as has been traditionally said [1], and hedonic responses strongly urge people to consume food [2]. Therefore, elucidating the physiological correlates of subjective hedonic experiences to visual food stimuli is of theoretical and practical interest. As several researchers [3,4,5] have proposed in the general literature on emotion, physiological activity may be the underlying mechanism of subjective hedonic experiences while observing food. Physiological measures are objective indices of subjective hedonic reactions to food materials, which food product developers use to develop new food materials complementing subjective hedonic ratings.

A few previous studies have reported that facial electromyography (EMG) signals were associated with subjective hedonic ratings while viewing food images [6,7,8]. Specifically, Greenwald et al. [6] assessed ratings of valence and arousal (i.e., the quality and intensity of emotional experience, respectively) and measured facial EMG of the corrugator supercilii muscle (related to frowning) and the zygomatic major muscle (related to smiling) while participants observed multiple images, including a food image. The valence ratings were negatively and positively associated with corrugator supercilii and zygomatic major EMG activity, respectively. Nath et al. [7,8] also measured facial EMG signals from these and additional muscles and assessed liking ratings while participants viewed several different food images. The liking ratings were negatively and positively associated with EMG activity of the corrugator supercilii [7,8] and zygomatic major muscles [7], respectively. Corroborating these data, other studies recorded EMG from these facial muscles while participants consumed liquid [9,10,11] or solid [12] food and showed similar associations between hedonic ratings, such as liking, wanting, and valence, and facial EMG activity of these muscles. Relatedly, some studies reported an association between facial expressions analyzed from video data and liking ratings during liquid food consumption [13,14]. Other studies found facial reactions using EMG or video data during the viewing [15,16] or consuming [17,18,19,20,21,22,23,24,25] of food stimuli. These findings suggest that subjective hedonic experiences while viewing food images may be associated with facial EMG activity.

However, whether facial EMG activity primarily reflects subjective hedonic experiences to food or objective nutritional values of the food remains unknown. It is generally assumed that foods with high caloric and/or fat contents are preferable [26] and that human hedonic responses may have evolved to select energy-dense and fatty foods for survival [27]. Consistent with this notion, a recent study has shown that subjective hedonic ratings, including liking, wanting, and valence, for food images were positively associated with the objective nutritional values of the caloric and fat contents of the food [28]. These data suggest that nutritional information about food could be a potential extraneous factor affecting the relationship between hedonic experience and physiological responses, including facial muscle activity. However, ample evidence has shown that hedonic responses to food reflect multiple factors, including learning experiences, and are not directly associated with the nutritional values of food [29]. Based on this line of evidence, we hypothesized that subjective hedonic responses to food images would have a stronger association with facial EMG activity than would the objective nutritional values of the food.

To test this hypothesis, we measured hedonic ratings and physiological signals while participants viewed food images. We presented photographs of Japanese food samples (Figure 1) that were validated as natural and hedonically evocative stimuli in a previous study of Japanese healthy adults [28]. These pictures were accompanied by objective nutrition information for each item, including caloric, carbohydrate, fat, and protein contents. Participants observed the food images and then rated their levels of liking, wanting, valence, and arousal. For physiological signals, we recorded facial EMG signals of the corrugator supercilii and zygomatic major muscles as measures of interest. We additionally recorded EMG signals of the masseter and suprahyoid muscles, which are related to food consumption [30,31,32], in an exploratory manner. As exploratory data, we also recorded skin potential response (SPR) and heart rate (HR), which reflect activity in the autonomic nervous system and are often related to emotional arousal [33,34,35]. We calculated intra-individual correlations [36,37] between the subjective ratings and physiological activity and evaluated the correlation coefficients in second-stage group analyses. Based on ample prior evidence, we predicted that liking, wanting, and valence ratings would show negative and positive associations with corrugator supercilii and zygomatic major EMG activity, respectively. More importantly, we predicted that the objective nutritional values of the food would show less evident associations with facial EMG activity than would subjective hedonic ratings.

## 2. Materials and Methods

### 2.1. Participants

We recruited 34 Japanese healthy young-adult participants (20 females; mean ± standard deviation (SD) age, 23.0 ± 5.0 years). We determined sample size using an *a priori* power analysis according to the guideline [38]. We used G*Power v3.1.9.2 software [39] and aimed to compare intra-individual correlation coefficients of facial EMG activity with subjective hedonic ratings vs. with objective nutrition information using dependent *t*-tests (two-tailed) with an *α* level of 0.05, power of 0.80, and effect size Cohen’s *d* of 0.5 (strong). The results indicated that 34 participants were sufficient. The recruitment of participants was conducted through advertising for healthy experimental participants presenting at the Kyoto University facility, and each participant received 1000 Japanese yen book coupons. No participants reported having physical or psychiatric problems. Participants reported their height and weight and the data showed that their average body mass index (BMI) was normal (mean ± SD, 20.7 ± 2.3; range, 15.1–26.9 kg/m^2^). All participants were not aware of facial EMG recording and had fasted for more than 3 h prior to the experiment. We assessed each participant’s hunger level before the experiment using a Likert scale (5-point), which ranged from 1 (full) to 5 (hungry) and found that the majority of them were relatively hungry (mean ± SD, 2.9 ± 0.8). After explaining the procedures, all participants gave written informed consent. This study was approved by Ethics Committee of the Unit for Advanced Studies of the Human Mind, Kyoto University on 24 July 2018. The experiment was conducted in accordance with institutional ethical guidelines and the tenets of the Declaration of Helsinki.

### 2.2. Apparatus

Experimental control was conducted using Presentation software v14.9 (Neurobehavioral Systems, Berkeley, CA, USA) on a Windows computer (HP Z200 SFF, Hewlett–Packard Japan, Tokyo, Japan). Stimuli were represented on a 19-inch cathode ray tube monitor (HM903D-A, Iiyama, Tokyo, Japan) at the resolution of 1024 horizontal × 768 vertical pixels and screen refresh rate of 100 Hz.

### 2.3. Stimuli

We used 32 color photographs of Japanese food samples (e.g., tempura and grilled fish) developed in a previous study [28] (Figure 1). The original food samples were selected from a Japanese food sample database, the Syokuiku Satisfactory “à La Carte” Tray (SAT) system (Iwasaki, Osaka, Japan). The SAT system contains samples of contemporary Japanese foods that appear in typical Japanese home meals and look similar to actual food. Crucially, the SAT system provides precise nutrient information for each item, including calories, carbohydrate, fat, and protein, through a careful analysis of contained materials. A previous study has shown that the photographs of these Japanese food samples were rated as natural [28]. Three other photographs from the database were used only for practice. The stimulus size was 640 horizontal × 480 vertical pixels, corresponding to a visual angle of approximately 23.9° horizontally × 17.9° vertically.

### 2.4. Procedure

The experiment was run on an individual basis in a sound-proof shielded room (Science Cabin, Takahashi Kensetsu, Tokyo, Japan). Before the experiment, the participants were informed that the objective of the experiment was to record electric signals from the skin in response to food images. Participants spent approximately 10 min adapting to the experimental environment. After three practice trials, a total of 32 trials were presented in random order separated by a varying intertrial interval (20–30 s).

Each trial began with the fixation point (a white cross) presented for 1 s on the black screen. Then, a food image was presented for 5 s. After that, a response display with four rating scales was presented until the participants finished their responses. The participants were instructed to rate their experiences while viewing the image. Four 9-point rating scales were presented simultaneously in fixed order: liking, wanting, valence, and arousal. Liking and wanting were evaluated using lines with numbers and anchors at the ends with ranges from 1 (dislike) to 9 (like) and 1 (do not want to eat) to 9 (want to eat), respectively. Valence and arousal evaluated using numbers, anchors at the ends, and the schematic images of a self-assessment manikin [40].

### 2.5. Physiological Data Recording

Facial EMG signals of the corrugator supercilii, zygomatic major, masseter, and suprahyoid muscles were recorded using pre-gelled, self-adhesive silver/silver chloride (Ag/AgCl) electrodes with 0.7-cm diameter and 1.5-cm interelectrode distance (Prokidai, Sagara, Japan). The electrodes were attached according to guidelines [41,42] and methods in previous studies [12,31,32]. A ground electrode was set in the middle of the forehead. The EMG signals were amplified and digitally sampled at 1 kHz using an EMG-025 amplifier (Harada Electronic Industry, Sapporo, Japan) and PowerLab 16/35 data collection system and LabChart Pro v8.0 software (ADInstruments, Dunedin, New Zealand). Online filter was applied (20–400 Hz [43]). Unobtrusive videotaping was conducted using a web camera (HD1080P, Logicool, Tokyo, Japan).

SPR was recorded from the hypothenar eminence of the left palm and the left forearm using pre-gelled, self-adhesive Ag/AgCl electrodes with 1.0-cm diameter (Vitrode F, Nihonkoden, Tokyo, Japan). The electrodes were attached in accordance with guidelines [44]. SPR signals were amplified and digitally sampled using a SPN-01 amplifier (Skinos, Ueda, Japan) and the same data collection system and software as those used for the EMG recording except that there was no online filter.

HR was recorded using a photo-plethysmograph placed on the fingertip of the left second finger. The data were digitally sampled with the same data collection system and software as those used for the EMG recording except for no online filter. The software automatically calculated beats per minute.

### 2.6. Data Analysis

#### 2.6.1. Preprocessing

We analyzed facial EMG signals using Psychophysiological Analysis Software v3.3 (Computational Neuroscience Laboratory of the Salk Institute, La Jolla, CA, USA) and in-house programs run under Matlab 2018 (MathWorks, Natick, MA, USA). The EMG signals during the 0.5-s pre-stimulus baseline period immediately prior to the stimulus presentation (the fixation point) and the 5-s stimulus presentation period were extracted for each trial. A coder who was blind to the experimental conditions checked the video data and confirmed that no participants made large motion artifacts. The EMG signals for each trial were rectified, baseline-corrected, and averaged. The EMG signal values were then standardized for each participant. To remove the effect of outliers, data with values falling outside the range of ± 3 were removed.

The SPR and HR signals were analyzed in the same way as the EMG analysis except that the signals were not rectified.

#### 2.6.2. Statistical Analysis

We calculated Pearson’s correlation coefficients (*r*) for each participant to measure individual-level correlations among subjective ratings; between subjective ratings and objective nutritional information (calories (kcal) and relative caloric percentages of carbohydrate, fat, and protein); between subjective ratings and physiological activity; and between objective nutritional information and physiological activity. We predicted the associations between liking/wanting/valence and corrugator supercilii/zygomatic major EMG activity as described in the Introduction. Other relationships were analyzed in the same way for descriptive purposes. The intra-individual correlation coefficients were normalized using Fisher’s *z*-transformation and then analyzed with one-sample *t*-tests (two-tailed) to compare the means with zero, in accordance with previous studies that investigated subjective–physiological emotional concordance using food (e.g., [12]) and non-food (e.g., [45]) stimuli. Such a two-stage procedure has been shown to be valid as a random-effects analysis [46]. To visually illustrate the associations between subjective ratings and physiological activity at the group-level, we depicted the relationships between group-averaged subjective ratings and standardized physiological data. The differences between subjective–physiological and nutrition–physiological intra-individual correlation coefficients were also analyzed using paired *t*-tests (two-tailed) with the same two-stage procedure. The threshold for significance was set at *p* < 0.05.

Because some previous studies have suggested moderating effects of BMI and hunger on the hedonic [47,48] or nutritional [49,50] processing of food, we performed preliminary analyses of BMI and hunger. We calculated inter-individual correlation coefficients between these measures and intra-individual subjective–physiological or nutrition–physiological correlation coefficients. We only analyzed the physiological data of zygomatic major EMG, as the main analyses showed significant effects only for this measure. The results showed that neither BMI nor hunger level were significantly correlated with the correlation coefficients of zygomatic major EMG activity with subjective ratings or objective nutritional values (|*r*| < 0.21, *p* > 0.1). Hence, these participant factors were disregarded.

## 3. Results

### 3.1. Subjective Ratings

Table 1 shows the mean intra-individual correlation coefficients among hedonic subjective ratings (liking, wanting, valence, and arousal) for food images and between subjective ratings and objective nutritional values (calories, carbohydrate, fat, and protein). One-sample *t*-tests for the correlation coefficients after Fisher transformation showed that all subjective ratings were significantly and positively correlated with one another (*r* > 0.48, *p* < 0.001). In addition, all of the subjective ratings were positively correlated with objective caloric contents and %fat (*r* > 0.12, *p* < 0.05), whereas significant negative correlations were observed with %carbohydrate (*r* < −0.08, *p* < 0.05).

### 3.2. Association between Subjective Ratings and Physiological Activity

The association between subjective ratings and physiological activity while viewing food images at the individual-level was analyzed by calculating the intra-individual correlation coefficients between subjective ratings and the mean physiological activity during image presentation. One-sample *t*-tests after Fisher transformation demonstrated that the liking, wanting, and valence ratings were significantly and positively correlated with zygomatic major EMG activity (*r* > 0.10, *p* < 0.001; Figure 2 and Table 2). Figure 3 illustrates these associations using group-averaged data. Besides, a positive correlation between wanting ratings and SPR reached significance (*r* > 0.07, *p* < 0.05). No other associations between subjective ratings and physiological activity reached significance (*p* > 0.1).

### 3.3. Association between Objective Nutritional Information and Physiological Activity

The association between objective nutritional values and physiological activity at the individual-level was analyzed as in the above analyses for subjective ratings. No significant correlation was observed between objective nutritional values and physiological responses (*r* < 0.06, *p* > 0.1; Figure 4 and Table 3).

We analyzed the correlation coefficients using dependent t-tests to compare the degree of association between zygomatic major EMG activity and subjective hedonic ratings with that between zygomatic major EMG activity and objective nutritional values. The subjective ratings of liking, wanting, and valence were selected to test for associations with zygomatic major EMG activity and compared against all objective nutritional values. The results showed that all of these subjective ratings had significantly stronger correlations with zygomatic major EMG activity (*t* > 2.20, *p* < 0.05; Table 4).

## 4. Discussion

The subjective ratings of liking, wanting, and valence were positively associated with EMG activity in the zygomatic major muscle while the participants viewed food images. These results corroborate prior findings reporting positive associations between subjective valence/liking ratings and zygomatic major EMG activity in response to food images [6,7]. Our study did not show any evident association between hedonic ratings and corrugator supercilii EMG activity, which is inconsistent with previous studies [6,7,8]. These discrepant findings may be due to methodological differences across studies. For example, in contrast to previous studies, our stimuli did not include any negative stimuli, which may have resulted in a lack of activity in the corrugator supercilii muscle.

More importantly, our data newly revealed that zygomatic major EMG activity was not associated with objective nutritional values, including caloric, carbohydrate, fat, or protein contents. In addition, the results revealed a stronger association between zygomatic major EMG activity and subjective hedonic ratings than between zygomatic major EMG activity and objective nutritional values. These results could be important because subjective hedonic ratings of food images are positively associated with the nutritional value of food [28] and, hence, food nutritional values were candidate extraneous variables [51] explaining the link between hedonic responses and facial EMG activity. Our data indicate that facial EMG activity is a physiological correlate of subjective hedonic experiences but not of objective nutritional values associated with food images.

Our results have theoretical implications. First, although it is widely assumed that hedonic responses to food may have evolved to foster acquisition of energy-dense and fatty foods [27] and people are generally good at estimating the nutritional values of visual food stimuli [28,52,53], our data suggest that the hedonic appraisals that produce facial muscle activity are not based solely on such nutritional computation. The results are in line with empirically grounded theory suggesting that innate hedonic responses to food based on national values could be altered by learned values acquired through various experiences such as mere exposure, classical conditioning, and social learning [29]. Consistent with this proposal, numerous studies have shown that learning experiences modulated hedonic responses to food [54]. Behavioral genetic studies also showed that genetic factors were strongly influenced by learning experiences [55]. Taken together, these findings suggest that hedonic appraisal of visual food stimuli involves learned, non-nutritional values attributed to food and produces corresponding bodily responses, including facial expressions. Second, our data heighten the possibility that facial EMG activity is a physiological correlate of subjective hedonic responses to visual food stimuli. In the literature of general emotional processing, the association between subjective emotional experiences and physiological activity has long been debated, and several researchers have proposed that physiological activity may underlie subjective emotional experiences [3,4,5]. Collectively, our data suggest that the production of facial muscle activity and its interoceptive perception may play an important role in the production of subjective hedonic experiences during visual food processing.

Our results also have practical implications. Because hedonic reactions are important for eating behaviors [2], food product developers need to assess hedonic reactions to their new products. However, subjective ratings, which companies mainly rely on, can be biased [56]. Our results provide additional evidence that facial EMG can be an objective correlate of subjective hedonic experiences. Measuring facial EMG has disadvantages relative to assessing subjective ratings, such as the need for equipment and the process of attaching contacts on participants, but may also carry some unique advantages. First, unlike subjective ratings, EMG can be continuously recorded without interfering with behaviors. Second, some previous studies using non-food visual stimuli have shown that facial EMG could detect unconscious emotional responses [57,58,59], which may also occur in the case of visual processing of food [60,61,62,63]. Third, a recent human pharmacological study showed that administration of opioid antagonists reduced zygomatic major EMG activity during the consumption of liked liquid food, but did not modulate the subjective ratings of liking and wanting [11]. The data suggest that facial EMG may more directly reflect activity in the brain reward circuits than subjective ratings under some conditions. Collectively, these advantages suggest that facial EMG recording may provide unique information complementing subjective hedonic ratings in applied consumer research. It would be interesting to assess objective hedonic responses during the viewing of food product packaging in real situations by using wearable devices that measure facial EMG activity.

In addition to facial EMG activity, our results showed that SPR was positively associated with subjective wanting ratings for food images. These results suggest the possibility that autonomic nervous system activity could be associated with the subjective hedonic response to food images in addition to facial muscle activity. However, due to a scarcity of evidence, we could not make preplanned hypotheses regarding physiological measures other than facial EMG; hence, we investigated these measures in an exploratory manner by conducting multiple testing without correction [64]. Additional tests and further confirmatory evidence are needed regarding the association between subjective wanting ratings and autonomic nervous system activity in response to visual food stimuli.

There are several limitations to this study. First, our sample was small and lacked the statistical power to detect weaker effects. Future studies using a larger sample may reveal associations of other physiological measures with subjective hedonic ratings. Second, although our preliminary analyses showed no significant moderator effect of BMI on subjective–physiological or nutrition–physiological correlations, we assessed participants’ BMI only using self-report data, which could differ from measured BMI data [65]. In addition, most of our participants were in the normal weight range, and none were obese. Future investigations measuring BMI objectively and testing various weight groups may reveal an influence of BMI on the association between physiological activity and hedonic or nutritional processing of visual food stimuli. Third, we tested only healthy participants, so the effect of physical (e.g., diabetes) or mental (e.g., eating disorder) problems remain untested. Investigation of the present topic in clinical samples is an important matter for future research. Fourth, we did not assess participants’ psychological traits. Because several previous studies have reported that participant traits such as eating style modulate hedonic reactions to food images [61,66], further investigation is necessary to reveal the modulatory effects of such traits on subjective–physiological associations in hedonic food processing. Fifth, we tested food images only with Japanese participants. Although it is generally assumed that basic hedonic responses are universal [67], several studies have reported that hedonic reactions to visual food stimuli may be different across cultures [68,69,70,71]. Hence, the generalizability of the present findings to other cultures requires further investigation. Finally, we applied only linear analyses. As some recent studies have reported that non-linear analyses using artificial neural networks could effectively classify surface EMG signals [72], such analyses may more sensitively reveal associations between hedonic experiences and facial EMG activity during the observation of food images.

## 5. Conclusions

Our results demonstrated that facial EMG activity recorded from the zygomatic major muscle was positively correlated with ratings of liking, wanting, and valence, but not with any objective nutritional values. These data indicate that facial EMG activity in response to food images reflect not the objective nutritional values but the subjective hedonic experiences in response to the food. These data have theoretical implications that hedonic appraisals producing facial muscle activity are not based solely on nutritional computation, and facial muscle activity plays an important role in the production of subjective hedonic experiences during visual food processing. The data also have practical implications in that facial EMG recording may provide objective information complementing subjective hedonic ratings in applied consumer research.

## Figures and Tables

**Figure 1 nutrients-13-00011-f001:**
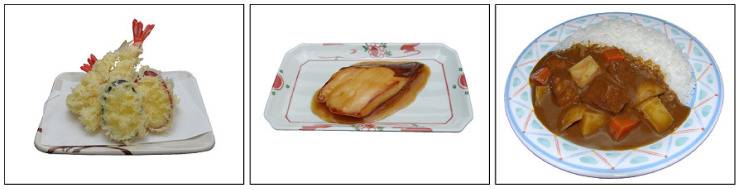
Examples of food images.

**Figure 2 nutrients-13-00011-f002:**
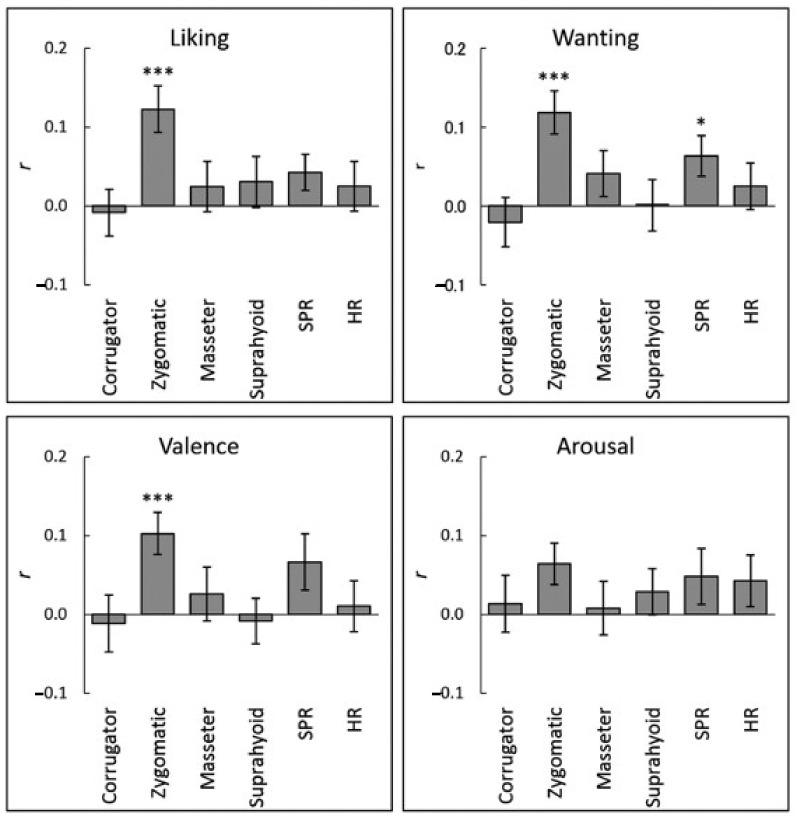
Mean (with standard error) intra-individual correlation coefficients (*r*) between subjective hedonic ratings and physiological activity while viewing food images. Corrugator = corrugator supercilii; Zygomatic = zygomatic major; SPR = skin potential response; HR = heart rate. *** *p* < 0.001; * *p* < 0.05.

**Figure 3 nutrients-13-00011-f003:**
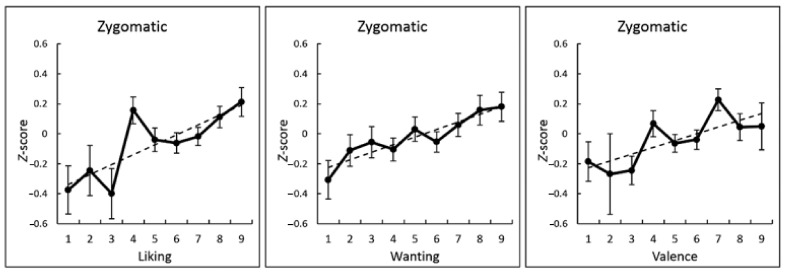
Group-mean (with standard error) values (dots) and regression lines (dashed lines) of subjective ratings (liking, wanting, and valence) and zygomatic major electromyography activity (standardized for each individual).

**Figure 4 nutrients-13-00011-f004:**
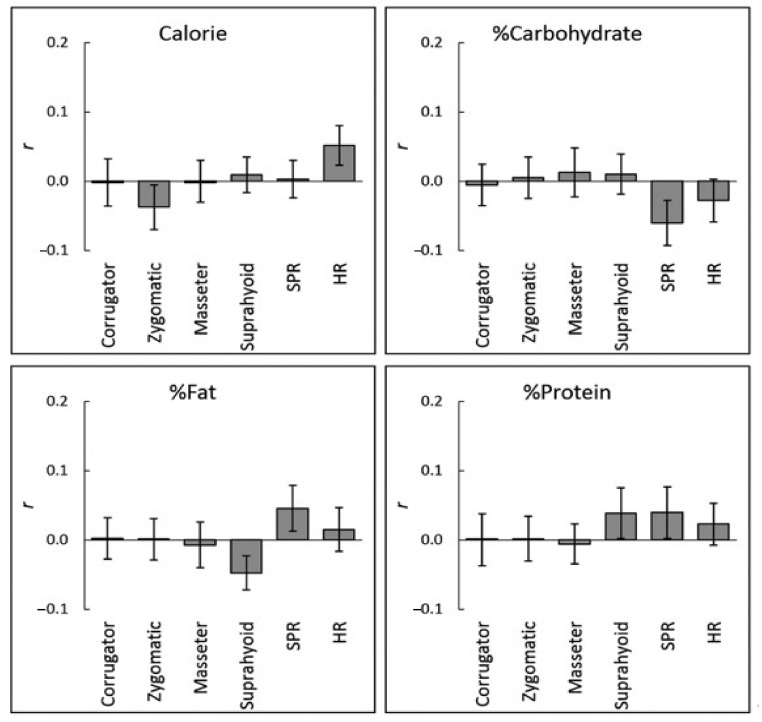
Mean (with standard error) intra-individual correlation coefficients (*r*) between objective nutritional values and physiological activity while viewing food images. No significant correlation was observed (*p* > 0.10). Corrugator = corrugator supercilii; Zygomatic = zygomatic major; SPR = skin potential response; HR = heart rate.

**Table 1 nutrients-13-00011-t001:** Mean (with standard error) correlation coefficients among subjective ratings and between subjective ratings and nutritional information values.

	Wanting	Valence	Arousal	Calorie	Carbohydrate	Fat	Protein
Liking	**0.77**	**0.66**	**0.51**	**0.27**	**−0.15**	**0.19**	−0.04
	**(0.03)**	**(0.04)**	**(0.06)**	**(0.04)**	**(0.03)**	**(0.03)**	(0.03)
Wanting		**0.73**	**0.57**	**0.19**	**−0.13**	**0.14**	−0.01
		**(0.04)**	**(0.05)**	**(0.04)**	**(0.04)**	**(0.04)**	(0.04)
Valence			**0.48**	**0.20**	**−0.11**	**0.13**	−0.04
			**(0.07)**	**(0.04)**	**(0.04)**	**(0.04)**	(0.04)
Arousal				**0.20**	**−0.09**	**0.13**	−0.06
				**(0.04)**	**(0.04)**	**(0.03)**	(0.04)

Significant results (*p* < 0.05) are in bold.

**Table 2 nutrients-13-00011-t002:** Results of one-sample *t*-tests (two-tailed; *t*-, *p*-, and Cohen’s *d*-values) for correlation coefficients between subjective hedonic ratings and physiological signals.

Rating	Statistic	Physiological					
		Corrugator	Zygomatic	Masseter	Suprahyoid	SPR	HR
Liking	*t*	0.29	**4.17**	0.76	0.95	1.84	0.79
	*p*	0.774	**0.000**	0.450	0.351	0.074	0.437
	*d*	0.05	**0.71**	0.12	0.16	0.31	0.13
Wanting	*t*	0.66	**4.32**	1.41	0.03	**2.44**	0.84
	*p*	0.515	**0.000**	0.169	0.980	**0.020**	0.408
	*d*	0.11	**0.74**	0.24	0.00	**0.41**	0.15
Valence	*t*	0.32	**3.84**	0.76	0.29	1.86	0.32
	*p*	0.751	**0.001**	0.451	0.773	0.072	0.753
	*d*	0.07	**0.66**	0.14	0.05	0.30	0.06
Arousal	*t*	0.49	1.81	0.24	0.85	1.44	1.28
	*p*	0.629	0.079	0.813	0.404	0.158	0.208
	*d*	0.08	0.31	0.04	0.15	0.25	0.22

Degrees of freedom are 33 for all. Significant results (*p* < 0.05) are in bold. SPR = skin potential response; HR = heart rate.

**Table 3 nutrients-13-00011-t003:** Results of one-sample *t*-tests (two-tailed; *t*-, *p*-, and Cohen’s *d*-values) for correlation coefficients between nutritional information and physiological signals.

Rating	Statistic	Physiological					
		Corrugator	Zygomatic	Masseter	Suprahyoid	SPR	HR
Liking	*t*	0.06	1.17	0.02	0.35	0.11	1.82
	*p*	0.955	0.249	0.986	0.727	0.914	0.077
	*d*	0.01	0.20	0.01	0.06	0.02	0.31
Wanting	*t*	0.18	0.17	0.35	0.35	1.87	0.90
	*p*	0.859	0.866	0.726	0.731	0.071	0.375
	*d*	0.03	0.03	0.05	0.06	0.32	0.16
Valence	*t*	0.08	0.03	0.22	1.95	1.37	0.48
	*p*	0.936	0.977	0.828	0.060	0.179	0.637
	*d*	0.01	0.00	0.04	0.33	0.23	0.09
Arousal	*t*	0.00	0.06	0.21	1.06	1.07	0.76
	*p*	0.998	0.956	0.838	0.296	0.294	0.453
	*d*	0.00	0.01	0.03	0.18	0.19	0.13

Degrees of freedom are 33 for all. No significant correlation was observed (*p* > 0.10). SPR = skin potential response; HR = heart rate.

**Table 4 nutrients-13-00011-t004:** Results of dependent *t*-tests (two-tailed; *t*-, *p*-, and Cohen’s *d*-values) contrasting correlation coefficients between zygomatic major muscle activity and hedonic ratings with those between muscle activity and nutritional information.

Rating	Statistic	Nutrition			
		Calorie	Carbohydrate	Fat	Protein
Liking	*t*	**4.88**	**2.98**	**2.89**	**2.56**
	*p*	**0.000**	**0.005**	**0.007**	**0.015**
	*d*	**0.89**	**0.68**	**0.70**	**0.67**
Wanting	*t*	**4.28**	**2.95**	**2.77**	**2.68**
	*p*	**0.000**	**0.006**	**0.009**	**0.011**
	*d*	**0.89**	**0.68**	**0.70**	**0.67**
Valence	*t*	**4.00**	**2.21**	**2.92**	**2.41**
	*p*	**0.000**	**0.034**	**0.006**	**0.022**
	*d*	**0.81**	**0.59**	**0.62**	**0.58**

Degrees of freedom are 33 for all. Significant results (*p* < 0.05) are in bold.

## Data Availability

The data presented in this study are available on request from the corresponding author.

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
