# Peer review of "Facial EMG Activity Is Associated with Hedonic Experiences but Not Nutritional Values While Viewing Food Images"

_nutrients, 2020, doi:10.3390/nu13010011_

Round 1
Reviewer 1 Report
I had the opportunity to review the manuscript at hand. I recommend the authors to revise the following points:
Introduction:
- EMG may be related to *subjective* ratings of nutritional values (i.e., calories) rather than to *objective* nutritional values as it was shown that subjective ratings of calorie content can be related to body-mass-index (BMI). I.e., individuals with overweight or obesity tent to underestimate the calorie content of foods (e.g., Larkin & Martin, 2016). Thus, EMG might relate to objective nutritional values if accounting for differences in BMI.
Method:
- Participants: Does ‘healthy’ mean that the participants had no (psychological) disorders, or does it also mean that only participants with normal weight were included? As mentioned above BMI might represent an important covariate. Thus, the sample mean and range of BMI would add valuable information to the characterization of the sample (if BMI data were assessed).
Results:
- I am a bit skeptical about the results of this study due to the small sample size. As a strong effect size (d = 0.5) was assumed in the estimation of the sample size, smaller effects may remained undetected. Thus, further studies might be needed to clarify whether small to medium effects were undetected.
- Additionally, the number of t-tests used might contribute to an accumulation of α-errors. Only the analysis regarding corrugator supercilii and zygomatic major were hypothesis driven, all other measures ‘were analyzed […] for descriptive purpose’ only)
Discussion:
- Could you discuss the limitations of your sample concerning the generalizability to individuals with broader BMI range and eating disorders?
Author Response
Thank you for your helpful suggestions for improving our manuscript, which we have modified accordingly. Additionally, we have modified several descriptions in accordance with the Editor’s guidance. Major changes to the manuscript are shown in red text. Language-related changes have also been made by a professional English-language editing service (http://www.textcheck.com/certificate/lw0d8G); these changes are not highlighted unless they have altered the content.
Point 1
Introduction:
- EMG may be related to *subjective* ratings of nutritional values (i.e., calories) rather than to *objective* nutritional values as it was shown that subjective ratings of calorie content can be related to body-mass-index (BMI). I.e., individuals with overweight or obesity tent to underestimate the calorie content of foods (e.g., Larkin & Martin, 2016). Thus, EMG might relate to objective nutritional values if accounting for differences in BMI.
Response
Because we assessed participants’ self-reported BMI, we have explained this process in the Materials and Methods section (p. 3). We also calculated inter-individual correlation coefficients between BMI and intra-individual subjective–physiological or nutrition–physiological correlation coefficients and found no significant effects. We have reported these preliminary results in the Materials and Methods section (p. 4). However, we acknowledge that we did not measure participants’ BMI directly, so we have noted this issue as a limitation of the study in the Discussion section (p. 10).
Point 2
Method:
- Participants: Does ‘healthy’ mean that the participants had no (psychological) disorders, or does it also mean that only participants with normal weight were included? As mentioned above BMI might represent an important covariate. Thus, the sample mean and range of BMI would add valuable information to the characterization of the sample (if BMI data were assessed).
Response
We described the participants as healthy, because we recruited them by advertising for healthy experimental participants, and all of the participants reported having no physical or psychiatric problems. We have clarified this in the Materials and Methods section (p. 3). As we employed participants’ self-reported BMI, we have also explained this in the same section.
Point 3
Results:
- I am a bit skeptical about the results of this study due to the small sample size. As a strong effect size (d = 0.5) was assumed in the estimation of the sample size, smaller effects may remained undetected. Thus, further studies might be needed to clarify whether small to medium effects were undetected.
Response
As suggested, we have discussed our small sample size as a limitation of this study in the Discussion section (p. 10)
Point 4
- Additionally, the number of t-tests used might contribute to an accumulation of α-errors. Only the analysis regarding corrugator supercilii and zygomatic major were hypothesis driven, all other measures ‘were analyzed […] for descriptive purpose’ only)
Response
As suggested, in the Introduction section (p. 2), we have explicitly clarified that our measurement of physiological signals other than facial EMG was exploratory. In the Discussion section (p. 9), we have emphasized that the analyses of these measures were conducted for descriptive purposes and that further studies are needed to confirm the findings.
Point 5
Discussion:
- Could you discuss the limitations of your sample concerning the generalizability to individuals with broader BMI range and eating disorders?
Response
In response to this advice, we have noted in the Discussion section (p. 10) that testing only non-obese and mentally healthy participants was a limitation of this study.
Reviewer 2 Report
Dear Authors,
The Authors presented very interesting results of research. This area of research is relatively new and not many papers are available. Unfortunately, in its current state, the manuscript is need to be improve.
Comments and Suggestions for Authors
Introduction
In my opinion the Introduction is not well prepared. Information from lines 40-53 is rather Discussion of results, and from lines 63-78 is rather section Materials and Methods.
The introduction does not demonstrate a gap in the literature on why this research should be conducted.
What are the practical and theoretical implications of the research?
I found no clear starting hypothesis for the present study and their results more than predictable. What is a main difference between Your previous article published in this year in Nutrients Journal. What's new in this manuscript, besides the food type? (Wataru Sato, Kazusa Minemoto, Akira Ikegami, Makoto Nakauma, Takahiro Funami, Tohru FushikiFacial EMG Correlates of Subjective Hedonic Responses During Food Consumption, Nutrients 2020, 12(4), 1174; https://doi.org/10.3390/nu12041174).
I would advise the authors to look at their aim and revise the manuscript
Methodology:
No details about the nutrition value of presented for consumers food products.
Not sure what is meant by "valence" in this manuscript as this does not reflect in the article.
Why only 34 people were recorded by Facial EMG?
This section is not clear at all. How respondents described nutrition value of foods products?
Did they even described nutrition value based on picture or only liking?
Authors should added details about experiment, because only EMG is well described. We don't know that only hedonic scales was used, maybe additional questions were used.
Conclusions
Conclusions are logical, but a little bit enigmatic. Value can be added if you make your conclusions more applicable and presented their practical implication.
References
Authors should be careful to link the bibliography to their results and their hypotheses, many references seem to old.
In the references list, was used a literature sources that in my opinion should be changed by to a newer one.
Authors should be complete the lack of information about some references e.g. in 12 position of reference.
In the manuscript was used 35 references. Seventeen of them come from 2010-2020 years, four come from the 2000-2009 year, fourteen of them is from before 2000. Are these pre-2000 sources needed? Can they be replaced with newer ones? The reference needs to be improved.
I have tried to find an article in MDPI Journal, Appetite Journal, and Food Quality and Preference. There is a few articles but presented results in the other context.
I found the following articles, maybe will replace old positions in References, but in not all authors emotions is measured by Face EMG. A few articles is connected with Face Reader, other method of measuring emotions.
- Mads Jochumsen, Imran Khan Niazi, Muhammad Zia ur Rehman, Imran Amjad, Muhammad Shafique, Syed Omer Gilani and Asim Waris, Decoding Attempted Hand Movements in Stroke Patients Using Surface Electromyography, Sensors 2020, 20(23), 6763; https://doi.org/10.3390/s20236763 - 26 Nov 2020.
- LEONARD H. EPSTEIN, ROCCO A. PALUCH, Habituation of Facial Muscle Responses to Repeated Food Stimuli, Appetite, Volume 29, Issue 2, 1997, pp. 213-224, https://doi.org/10.1006/appe.1997.0102.
- Fuentes, S.; Gonzalez Viejo, C.; Torrico, D.D.; Dunshea, F.R. Development of a Biosensory Computer Application to Assess Physiological and Emotional Responses from Sensory Panelists. Sensors 2018, 18, 2958.
- Elizabeth C. Nath, Peter R. Cannon, Michael C. Philipp, Co-acting strangers but not friends influence subjective liking and facial affective responses to food stimuli, Food Quality and Preference, Volume 82, 2020, 103865, https://doi.org/10.1016/j.foodqual.2019.103865.
- Yasmin Ioannides, John Seers, Marianne Defernez, Carol Raithatha, M. Scott Howarth, Andrew Smith, E. Kate Kemsley, Electromyography of the masticatory muscles can detect variation in the mechanical and sensory properties of apples, Food Quality and Preference, Volume 20, Issue 3, 2009, pp. 203-215, https://doi.org/10.1016/j.foodqual.2008.09.007.
- Torrico et al. 2018: Images and chocolate stimuli affect physiological and affective responses of consumers: A cross-cultural study, Food Quality and Preference, 65, 60-71.
- de Wijk et al. (2014). ANS responses and facial expressions differentiate between the taste of commercial breakfast drinks. PLoS One, 9 (4), e93823
Reviewer
Author Response
Thank you for your helpful suggestions for improving our manuscript, which we have modified accordingly. Additionally, we have modified several descriptions in accordance with the Editor’s guidance. Major changes to the manuscript are shown in red text. Language-related changes have also been made by a professional English-language editing service (http://www.textcheck.com/certificate/lw0d8G); these changes are not highlighted unless they have altered the content.
Point 1
Introduction
In my opinion the Introduction is not well prepared. Information from lines 40-53 is rather Discussion of results, and from lines 63-78 is rather section Materials and Methods.
Response
Following your advice, we have improved the second and final paragraphs of the Introduction section. For the former, we conducted thorough literature review. For the latter, we deleted some information that was too specific (e.g., the number of stimuli) and added information necessary to understanding the present study.
Point 2
The introduction does not demonstrate a gap in the literature on why this research should be conducted.
What are the practical and theoretical implications of the research?
Response
We believe that the previous literature lacked an investigation of the effects of nutritional value on the relationship between subjective hedonic experience and facial EMG responses. We have explained this deficit in the Introduction section (p. 2). Our results clearly showed that facial EMG responses during the observation of food images reflects the hedonic, but not the nutritional, values of food. We have discussed the practical and theoretical implications of these findings in the Discussion section (p. 9).
Point 3
I found no clear starting hypothesis for the present study and their results more than predictable.
Response
As suggested, we have described our hypothesis in the Introduction (p. 2).
Point 4
What is a main difference between Your previous article published in this year in Nutrients Journal. What's new in this manuscript, besides the food type? (Wataru Sato, Kazusa Minemoto, Akira Ikegami, Makoto Nakauma, Takahiro Funami, Tohru FushikiFacial EMG Correlates of Subjective Hedonic Responses During Food Consumption, Nutrients 2020, 12(4), 1174; https://doi.org/10.3390/nu12041174).
I would advise the authors to look at their aim and revise the manuscript
Response
The main differences between this study and our previous study were: (1) this study tested visual processing of food in contrast to consumption processing, and (2) this study newly revealed that facial EMG responses to food stimuli were associated not with the nutritional but with the hedonic value of food stimuli. To clarify this information, we have carefully described these different forms of processing during our review of the previous literature in the Introduction (p. 2). Furthermore, we have thoroughly discussed the importance of investigating the relationship between facial EMG activity and the nutritional value of food stimuli in the Introduction section (p. 2).
Point 5
Methodology:
No details about the nutrition value of presented for consumers food products.
Response
As suggested, we have added details regarding the nutritional value of food items in the Materials and Methods section (p. 3).
Point 6
Not sure what is meant by "valence" in this manuscript as this does not reflect in the article.
Responses
As suggested, in the Introduction section (p. 1), we have clarified that the term “valence” in this article refers to the quality of emotional experience.
Point 7
Why only 34 people were recorded by Facial EMG?
Responses
We determined our sample size of 34 people based on an a priori power analysis following the guidelines of Kyonka (2018: Perspect Behav Sci, 42, 133-152). We have provided this information in the Materials and Methods section (p. 2). However, because the sample size was not large enough to detect weak effects, we have discussed this as one of the study limitations in the Discussion section (p. 10).
Point 8
This section is not clear at all. How respondents described nutrition value of foods products?
Did they even described nutrition value based on picture or only liking?
Response
Instead of assessing the subjective nutritional values of food products, we used the foods’ objective nutrition values provided by the developer of the materials. We have clarified this in the Introduction (p. 2) and Materials and Methods (p. 3) sections.
Point 9
Authors should added details about experiment, because only EMG is well described. We don't know that only hedonic scales was used, maybe additional questions were used.
Response
As suggested, we added the information about other measures. Because we assessed participants’ BMI, we have described this information in the Materials and Methods section (p. 3). As we did not assess other variables, we have noted this and have discussed this as a limitation of the study in the Discussion section (p. 10).
Point 10
Conclusions
Conclusions are logical, but a little bit enigmatic. Value can be added if you make your conclusions more applicable and presented their practical implication.
Response
As suggested, we have added comments on the study’s implications in the conclusions (p. 11).
Point 11
References
Authors should be careful to link the bibliography to their results and their hypotheses, many references seem to old.
In the references list, was used a literature sources that in my opinion should be changed by to a newer one.
In the manuscript was used 35 references. Seventeen of them come from 2010-2020 years, four come from the 2000-2009 year, fourteen of them is from before 2000. Are these pre-2000 sources needed? Can they be replaced with newer ones? The reference needs to be improved.
Response
We appreciate your advice and have updated the references accordingly.
Point 12
Authors should be complete the lack of information about some references e.g. in 12 position of reference.
Response
As suggested, we have added missing information to the references.
Reviewer 3 Report
This is a well written and interesting study. I have two questions listed below for the authors.
Line 83: How were the participants recruited?
Line 264: Would this be practical for food companies based on the cost of the EMG recording?
Author Response
Thank you for your helpful suggestions for improving our manuscript, which we have modified accordingly. Additionally, we have modified several descriptions in accordance with the Editor’s guidance. Major changes to the manuscript are shown in red text. Language-related changes have also been made by a professional English-language editing service (http://www.textcheck.com/certificate/lw0d8G); these changes are not highlighted unless they have altered the content.
Point 1
Line 83: How were the participants recruited?
Response
Participants were recruited through advertisements for experimental participants. We have described this process in the Materials and Methods section (p. 3)
Point 2
Line 264: Would this be practical for food companies based on the cost of the EMG recording?
Response
We agree that recording facial EMG has disadvantages relative to subjective ratings that could reduce the practical significance of our results. However, the data suggest some possible advantages of facial EMG, including continuous recording, detection of subtle emotional responses, and strong relationships with brain reward circuit activity. We have discussed these disadvantages and advantages of facial EMG in the Discussion section (p. 9).
Round 2
Reviewer 2 Report
The authors have changed many parts of the planned paper according to my suggestions.
The Section Introduction, Material, and Methods, Limitation of study, Conclusion, as well as References were corrected.
In the new version (v.2) of the manuscript I noticed subsection numbering is missing for example in line 198 there is - Subjective ratings, but in my opinion, should be 3.1. Subjective ratings.
W section References, when authors have listed names of authors in a few places are comma instead of a semicolon.
I would like to thank the authors for considering my comments and applaud them for the major revisions to improve their manuscript.
The manuscript is interesting and valuable. I recommend the manuscript to be published in this form.
Reviewer